# A recombinant BBSome core complex and how it interacts with ciliary cargo

Björn Udo Klink[1,2], Eldar Zent[2], Puneet Juneja[1], Anne Kuhlee[1], Stefan Raunser[1]*, Alfred Wittinghofer[2]*

[1]Department of Structural Biochemistry, Max Planck Institute of Molecular Physiology, Dortmund, Germany; [2]Structural Biology Group, Max Planck Institute of Molecular Physiology, Dortmund, Germany

**Abstract** Cilia are small, antenna-like structures on the surface of eukaryotic cells that harbor a unique set of sensory proteins, including GPCRs and other membrane proteins. The transport of these proteins involves the BBSome, an eight-membered protein complex that is recruited to ciliary membranes by the G-protein Arl6. BBSome malfunction leads to Bardet-Biedl syndrome, a ciliopathy with severe consequences. Short ciliary targeting sequences (CTS) have been identified that trigger the transport of ciliary proteins. However, mechanistic studies that relate ciliary targeting to BBSome binding are missing. Here we used heterologously expressed BBSome subcomplexes to analyze the complex architecture and to investigate the binding of GPCRs and other receptors to the BBSome. A stable heterohexameric complex was identified that binds to GPCRs with interactions that only partially overlap with previously described CTS, indicating a more complex recognition than anticipated. Arl6•GTP does not affect these interactions, suggesting no direct involvement in cargo loading/unloading.
DOI: https://doi.org/10.7554/eLife.27434.001

*For correspondence:
stefan.raunser@mpi-dortmund.
mpg.de (SR);
alfred.wittinghofer@mpi-
dortmund.mpg.de (AW)

Competing interests: The authors declare that no competing interests exist.

## Introduction

Bardet-Biedl syndrome (BBS) is a pleiotropic genetic disease (*Beales et al., 1997*; *Zaghloul and Katsanis, 2009*) characterized by obesity, polydactyly, renal dystrophy and cystic kidneys. These phenotypes are associated with malfunctions of cilia, small antenna-like structures on almost every cell, which perform important sensory and regulatory functions. Up to now more than 19 genes have been found mutated in BBS which account for about 80% of the clinically examined cases (*Khan et al., 2016*). Of these, eight BBS gene products were found to form a stable complex called the BBSome. BBS1 is part of this complex, and has by far the highest mutational load, contributing to about a quarter of all BBS mutations in Europe and North America (*Khan et al., 2016*; *Forsythe and Beales, 2013*). The BBSome was first identified in a pulldown experiment with BBS4 as a bait (*Nachury et al., 2007*), which forms a stable complex with the six BBSome proteins BBS1,2,5,7,8 and 9. BBS18, also known as BBIP10, was later identified as another integral part of the BBSome complex (*Loktev et al., 2008*). The formation of the BBSome complex has been postulated to additionally require the three chaperonine-like BBS proteins BBS6, 10 and 12. These proteins act together with the eukaryotic CCT complex and are described to form a BBS/CCT complex as a starting point for the sequential assembly of the BBSome (*Seo et al., 2010*; *Zhang et al., 2012b*). BBSome subunits show a domain architecture similar to proteins from vesicle-forming complexes like COPI/COPII and clathrin (*Jin et al., 2010*), and from the intraflagellar transport complex (IFT) (*van Dam et al., 2013*). The membrane association of the BBSome was found to be regulated by the Arf-like small G protein Arl6/BBS3, in analogy to the Arf subfamily proteins Arf1 and Sar1 in COPI/II and clathrin coats (*Cherfils, 2014*; *Wieland and Harter, 1999*). BBS3 in the GTP-bound conformation binds to the N-terminal ß-propeller domain of BBS1 (*Mourão et al., 2014*), thereby

recruiting patches of BBSome particles, as was demonstrated on synthetic liposomes (*Jin et al., 2010*). While COP complexes bud vesicles by forming cage-like structures on membranes (*Lee and Goldberg, 2010*), the BBS3-induced BBSome patches seem to be planar.

The formation of a planar coat resembles the IFT complex, which also has evolved to form a specialized type of planar membrane coat (so-called IFT trains), and is required for transport within the cilium. The BBSome is attached to this complex, regulates assembly and stability of IFT trains, and travels together with it along the ciliary axoneme (*Wei et al., 2012*; *Williams et al., 2014*; *Pan et al., 2006*). However, being a relatively recent evolutionary development (*van Dam et al., 2013*), the details of BBSome function seem to have evolved differently in different organisms: While in *Chlamydomonas* the BBSome is sub-stoichiometric to IFT particles and does not influence IFT function or stability (*Lechtreck et al., 2009*), disruption of BBSome proteins in *C. elegans* destabilizes the entire IFT complex and affects the cohesion between IFT-A and IFT-B (*Wei et al., 2012*; *Pan et al., 2006*). A 1:1 stoichiometry of BBSome and IFT has been reported for mammalian olfactory sensory neurons (*Williams et al., 2014*).

In *Drosophila*, which has only a limited number of ciliated cells, the BBSome subunits BBS2 and BBS7 are completely absent. Their absence begs the intriguing question of the existence and functional relevance of BBSome subcomplexes. Downregulation of the transition zone protein NPHP5 was found to specifically reduce the ciliary localization of BBS2 and BBS5, while other BBSome subunits (BBS1,4,7,8,9 and 18) were still targeted to cilia (*Barbelanne et al., 2015*). Consistently, by determining the spatial distribution of BBSome subunits in the retina it was observed that some BBSome subunits are differentially localized to different compartments of photoreceptor cilia (Wolfrum, Spitzbarth, private notification). These studies show that subcomplexes of the BBSome can exist and may have functional relevance.

The BBSome is believed to be responsible for the selection of ciliary transmembrane proteins for IFT-mediated transport (*Berbari et al., 2008b*). The hypothesis that the BBSome is mainly an import adapter that regulates ciliary import of cargo proteins (*Berbari et al., 2008b*; *Jin et al., 2010*; *Loktev and Jackson, 2013*) is questioned by studies that indicate a more crucial role in ciliary export (*Domire et al., 2011*; *Eguether et al., 2014*; *Liew et al., 2014*; *Xu et al., 2015*; *Lechtreck et al., 2013*; *Lechtreck et al., 2009*). As was demonstrated recently, malfunction of BBSome-mediated export can be compensated by ectocytosis of membranous buds at the tips of cilia where GPCRs and other cargo proteins get concentrated. This can lead to a counter-intuitive net loss of proteins that no longer get exported from cilia due to defects in the BBSome (*Nager et al., 2017*).

Relatively short peptide sequences, so-called ciliary targeting sequences (CTS) have been identified as factors modulating ciliary localization. However, no common denominator has been identified and a direct physical interaction between the BBsome and the CTS was only proven in a limited number of cases. The question thus arises how the BBSome recognizes its cargo. One of the best-studied examples of a cilium-localized receptor protein is the somastatin receptor 3 (SSTR3), which interacts with the BBSome via its third intracellular loop (3ICL) (*Jin et al., 2010*; *Berbari et al., 2008a*; *Berbari et al., 2008b*). Studies on chimeric receptors indicate that even though this interaction is sufficient to target non-ciliary receptors to the cilium, it´s mutation does not prevent SSTR3 ciliary targeting (*Berbari et al., 2008a*). This indicates that SSTR3 utilizes a rather complex signal for ciliary localization, with more than one epitope being important. It was also shown that the localization of SSTR3 does not only depend on the BBSome, but also on other factors, such as the ciliary import adapter protein Tulp3, which regulates the trafficking of many, but not all, integral membrane cargo to the cilium (*Badgandi et al., 2017*).

The BBSome subunit BBS1 emerged to be particularly important for cargo recognition, e.g. forming interactions with the C-terminal cytosolic tails of Smoothened (Smo) and Patched (Ptch1) from the hedgehog pathway (*Zhang et al., 2012a*; *Ruat et al., 2012*; *Bhogaraju et al., 2013*) and of the Leptin receptor (*Seo et al., 2009*) and Polycystin 1 (*Su et al., 2014*). However, additional BBSome subunits also contribute to cargo binding: For example, the C-terminal tail of Smoothened (Smo) was shown to form interactions with BBS1,4,5 and 7 (*Seo et al., 2011*), and a GST-tagged 3ICL of SSTR3 interacts, with varying affinities, with all BBSome subunits (*Jin et al., 2010*).

The molecular mechanisms that lead to ciliary localization are still not well understood. To gain a deeper insight into the functional role of the BBSome on a molecular level, we established the recombinant expression and purification of individual BBSome proteins, BBSome subcomplexes and the fully assembled BBSome in the baculovirus/insect cell system. While the fully assembled human

BBSome, at least when overexpressed in insect cells, is barely soluble, a subcomplex consisting of BBS1,4,5,8,9 and 18 is a highly stable monomer suitable for biochemical, biophysical and structural studies. The negative-stain EM reconstruction of this 6mer BBSome core indicates a Y-shaped arrangement, and combined with biochemical data, it allows first suggestions about the complex architecture. We show that it retains the ability to bind to Arl6/BBS3 and selected ciliary cargo proteins. We find the binding mode to cargo proteins to be more complex than anticipated.

## Results

### Isolation of individual BBSome subunits

Structural and mechanistic studies on the BBSome are hampered by the scarcity of material obtained from natural sources such as bovine eye. We thus focused on recombinant expression of the human BBSome in the baculovirus/insect cell system. We first tested the feasibility of purifying single components of the BBSome, the small GTP-binding protein BBS3 (Arl6) for which it is an effector, and the chaperonin proteins BBS6, 10 and 12, assuming that the latter might be required for a functional assembly of the core complex. All proteins could be overexpressed (*Figure 1—figure supplement 1A*), but only BBS5, BBS9 and BBS18 were found to be soluble in significant amounts (*Figure 1—figure supplement 1B*). BBS1 was degraded to a smaller fragment representing its ß-propeller domain, which could be expressed and purified (residues 1–430). The other BBS subunits were only soluble in minute amounts and often co-purified with chaperones like HSP 70 (*Figure 1—figure supplement 1B*). Removal of these reduced the yield even further.

### Stabilizing interactions in binary BBSome sub-complexes

Since most BBS subunits clearly required additional stabilizing factors or an improved folding machinery, we next analyzed how the yields and purities are affected by co-expressing BBS subunits. For efficient dual affinity purification of the desired complexes, different combinations of affinity tags were tested. Purification of several stoichiometric complexes could be achieved, from which the complexes of BBS9 with BBS8 and of BBS18 with BBS4 showed the strongest improvement in solubility compared to the individual components. The co-expression tests also placed BBS9 as a central organizational component of the BBSome, forming stable binary interactions with BBS1, 2, 5 and 8 (*Figure 1*). The BBS chaperonins BBS6 and BBS12 also stabilized each other significantly when coexpressed, but coexpressing them with BBSome subunits did not improve yields and/or purity of the latter.

### BBS9 can be divided in two soluble domains with distinct subunit specificity

BBS9 can be divided into an N-terminal ß-propeller domain (residues 1–368) and a C-terminal part containing predicted GAE, platform and alpha-helical domains (residues 363–887) (*Jin et al., 2010*). To test whether they show different interaction patterns, the N- and C-terminal parts were coexpressed with different combinations of BBSome proteins and analyzed for the formation of stable binary complexes. We could isolate complexes of BBS5 with the N-terminal ß-propeller (*Figure 1—figure supplement 2* and of BBS1 with the C-terminal part. The complex with BBS5 was far more soluble than the complex with BBS1, which could only be isolated in small amounts.

There is evidence that the full BBSome is monomeric in solution (*Nachury et al., 2007*), which is surprising considering that BBS9 can dimerize (*Knockenhauer and Schwartz, 2015*). Consistent with the study of Knockenhauer et al. (*Knockenhauer and Schwartz, 2015*), we found the C-terminal part of BBS9 to elute as a dimeric species in gel filtration, while the N-terminal ß-propeller domain of BBS9 runs as a monomer (*Figure 1—figure supplement 2*). We find that full length BBS9 preserves dimerization at high concentrations, but the elution peak shifts to lower molecular weight at lower concentrations. The observation of a single peak indicates a fast equilibrium between monomer and dimer.

### Crucial subunit interactions to stabilize a larger BBSome complex

The binary coexpression studies provided valuable information about direct interactions, the domains involved and the efficient use of purification tags. Apart from the qualitative information,

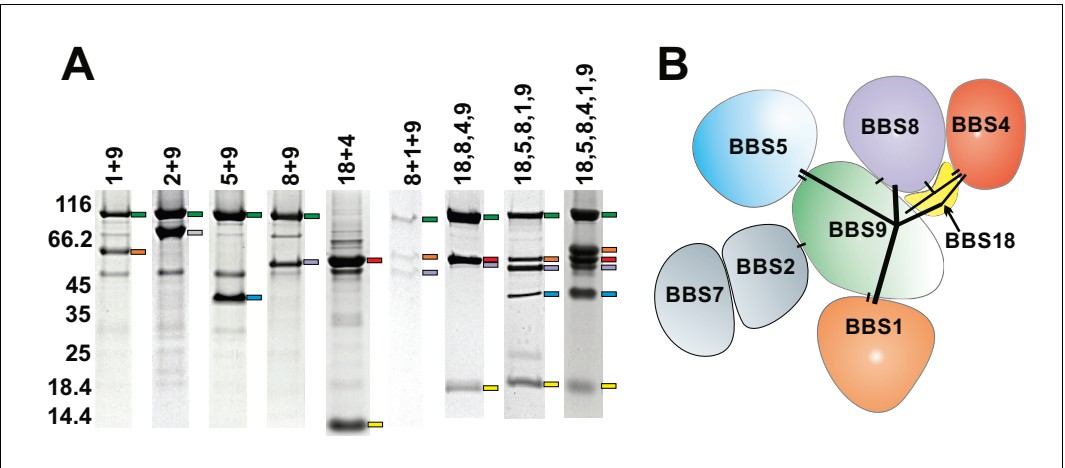

**Figure 1.** Subcomplexes of the BBSome. (**A**) Coomassie-stained SDS-PAGE of binary and multimeric BBSome complexes purified by Strep/Flag tandem affinity purification. The BBS proteins are indicated with color-coded marks. Variances in gel mobility of the individual components is caused by different affinity tags, as is most pronounced for BBS18. (**B**) Schematic subunit organization model that is consistent with expression and stability of subcomplexes (Components of the most stable core complex in color, and the two missing subunits in grey).
DOI: https://doi.org/10.7554/eLife.27434.002

The following figure supplements are available for figure 1:

**Figure supplement 1.** Coomassie-stained SDS-PAGE on individual BBSome subunits.
DOI: https://doi.org/10.7554/eLife.27434.003
**Figure supplement 2.** Normalized gel filtration profiles of different BBS complexes.
DOI: https://doi.org/10.7554/eLife.27434.004
**Figure supplement 3.** Purification of BBSome complexes.
DOI: https://doi.org/10.7554/eLife.27434.005

they also provided a more quantitative view on the possible improvement of solubility achieved by combining subunits. For example, the complex of BBS9, 8 and 1 (*Figure 1*) was stable, but it´s solubility compared to the binary complexes BBS9 +8 and BBS9 +1 was not improved. BBS2 and BBS7, subunits that are known to bind to each other and to BBS9 (*Seo et al., 2010*), rendered most tested complexes even less soluble. The introduction of the 10 kDa protein BBS18 however dramatically improved the yield of soluble complexes, as in a complex with BBS1,5,8 and 9. We also found BBS18 to be crucial to link BBS4 to this complex, which is consistent with an interaction of BBS18 with both BBS4 and with BBS8,9. In *Figure 1*, some of the purified subcomplexes with 2–6 components are shown. For those with sufficient yield, we also analyzed their oligomerization state in gel filtration experiments (*Figure 1—figure supplement 2*). All tested proteins presented a monomeric elution profile, with an exception of the C-terminal part of BBS9.

## The core BBSome

Coexpression of the full ensemble of eight BBSome subunits yielded only small amounts of complex with substoichiometric amounts of BBS2 and BBS7 (*Figure 1—figure supplement 3A,B*). This full BBSome was difficult to purify and ran as oligomeric species on gel filtration. Instead we found that complexes containing BBS18 but not BBS2 and 7 had a greatly improved stability and do not oligomerize. The largest and most stable subcomplex is composed of BBS1, 4, 5, 8, 9 and 18. We consider this a BBSome core complex since it shows the most promising structural and functional properties, as it retains most of the functionality of the full BBSome machinery known so far. Of note, we did not observe any additional improvement of sample quality of BBSome subcomplexes or the full BBSome upon coexpression with any or all of the BBS chaperonins BBS6, 10 and 12.

The 6mer core BBSome could readily be purified in milligram quantities, and is soluble up to concentrations higher than 150 µM (>50 mg/ml). A single Strep affinity purification yielded almost pure complex with no obvious imbalance in the stoichiometry of components. The complex eluted as a single entity in all subsequent purification steps (*Figure 1—figure supplement 3E*), and was even

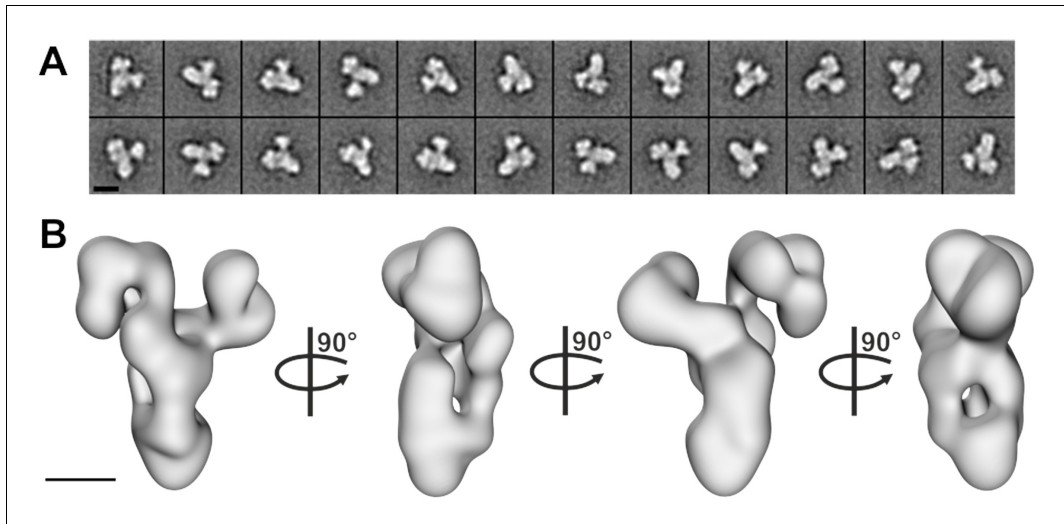

**Figure 2.** Negative stain electron microscopy analysis of the core BBSome complex. (**A–B**) Representative 2-D class averages (scale bar, 10 nm) (**A**) and a 3-D reconstruction (**B**) of the core BBSome (scale bar, 5 nm).
DOI: https://doi.org/10.7554/eLife.27434.006
The following figure supplement is available for figure 2:

**Figure supplement 1.** Negative stain EM of the core BBSome complex.
DOI: https://doi.org/10.7554/eLife.27434.007

resistant to high salt washes of up to 2.5 M NaCl, consistent with a previous study where the native BBSome was eluted with high salt concentrations from an Arl6 affinity column (*Jin et al., 2010*). The complex elutes in gel filtration with an apparent molecular weight of ~400 kDa, in acceptable agreement with the expected theoretical mass of 336.4 kDa (*Figure 1—figure supplement 3C,D*), but less consistent with it being a dimer.

## Electron microscopy of the core BBSome

To gain more information about the molecular arrangement of the 6mer core BBSome complex, we analyzed it by negative stain electron microscopy. The complex was intact, homogeneous and randomly oriented on the grid (*Figure 2—figure supplement 1A*) allowing us to calculate 2-D class averages (*Figure 2A*, *Figure 2—figure supplement 1C*) and a corresponding 3-D reconstruction at a resolution of ~23 Å (*Figure 2B*, *Figure 2—figure supplement 1B*). The complex has a distorted Y-shape (side view) with pseudo 3-fold symmetry and a loosely packed architecture, although we cannot identify the localization of individual subunits at this resolution.

## The core BBSome interacts with Arl6 (BBS3) and cargo proteins

From previous data, BBS1 seems to be the BBSome subunit most relevant for cargo recognition and Arl6 binding. We first tested the binding of Arl6 to the BBSome core complex. In pulldown experiments with immobilized Arl6, the purified 6mer BBSome core was captured in a nucleotide-dependent manner: While a 7.5 fold molar excess of Arl6•GppNHp precipitated the major part of the BBSome from solution, Arl6•GDP had no detectable affinity, which supports the notion that the BBsome is the effector of Arl6 (*Figure 3*). To test whether the core BBSome can recognize known BBSome cargo proteins, we generated a truncated (residues R546-L665) version of the Smo receptor, which contains a ciliary targeting motif predicted to interact with the BBSome (*Zhang et al., 2012a*; *Seo et al., 2011*; *Corbit et al., 2005*). In pulldown experiments as described above (*Figure 3*), the binding was clearly detectable, but considerably weaker than that of Arl6•GppNHp. In fluorescence polarization experiments (see below), we determined the affinity for a fluoresceine-labeled version of this Smo fragment to be 36.1 ± 3.9 μM.

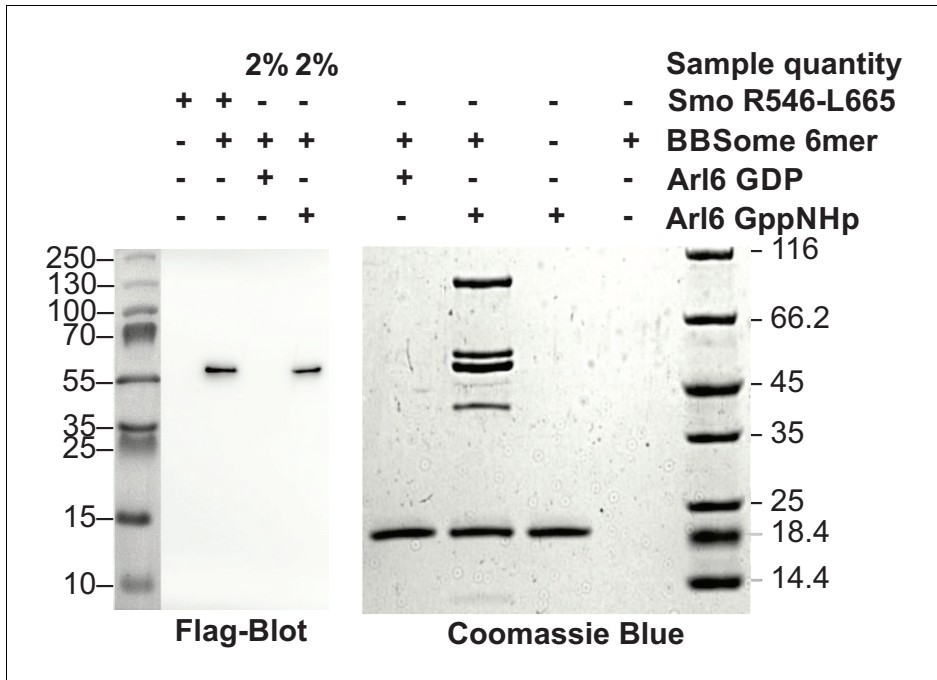

**Figure 3.** Pulldown of core BBSome with immobilized Arl6 and Smoothened domain R546-L665. Purified core BBSome samples were pulled down by a 7.5 fold molar excess of Arl6•GDP, Arl6•GppNHp or Smo(R546-L665) immobilized to Streptactin-coupled Sepharose beads. While no interaction of the BBSome with Arl6•GDP was detectable, Arl6•GppNHP captured most of the BBSome sample. Smo(R546-L665) bound weaker, but still captured >2% of the BBSome sample. This was approximated by a comparison of the 1:50 diluted Arl6•GppNHp elution with the undiluted Smo(R546-L665) elution in a Flag-blot against Flag-BBS8 from the BBSome core complex.

DOI: https://doi.org/10.7554/eLife.27434.008

## Identification of binding epitopes in BBSome cargo proteins

In order to analyze the binding mode of BBSome cargo proteins and to delineate a putative ciliary targeting signal in an unbiased approach, we switched to a high throughput method using high density peptide arrays (*Beyer et al., 2007*; *Stadler et al., 2008*). We hypothesized that the BBSome binding requires relatively short peptide motifs, in analogy to sequences recognized by COP complexes (*Mancias and Goldberg, 2008*; *Jackson et al., 2012*; *Mossessova et al., 2003*). Since GPCRs emerged as a major class of proteins transported by the BBSome (*Omori et al., 2015*; *Schou et al., 2015*; *Jin et al., 2010*), we tested several ciliary GPCRs (*Table 1*) and also other ciliary receptors for interaction with the core BBSome. For this, we analyzed an array of ~4000 15 mer peptides that represent a scan of the protein sequences, with a single residue offset from one peptide to the next. Such an assay is suitable for locating relatively short binding epitopes that do not depend on the context of the native proteins. It cannot be excluded however that the binding affinities might be modulated in the context of the native protein environment and that other binding sites might exist that cannot be identified by this method. *Figure 4—figure supplement 1* shows a full overview of the microarrays and *Figure 4* an exemplary profile along a part of the SSTR3 receptor: A quantification of the signal intensities from the microarray (*Figure 4C*) indicates that the core BBSome has the strongest interaction with peptides from the 3ICL and the C-terminal region of SSTR3, indicated in blue and green in the intensity diagram and in the context of the snake plot of SSTR3 (*Figure 4D*).

We found BBSome binding motifs in most of the GPCRs tested and in Polycystin-1, Polycystin-2, the leptin receptor and Fibrocystin (*Figure 4—figure supplement 1*). Significant BBSome-GPCR binding was almost exclusively observed for peptides from the 3ICL and the C-terminus, which are the regions previously described to interact with the BBSome and to contain ciliary targeting sequences like Ax(S/A)xQ (*Berbari et al., 2008a*) or (I/V)KARK (*Mukhopadhyay et al., 2013*) in the

**Table 1.** Interactions of GPCR peptides with the core BBSome.

| GPCR | Uniprot accession number | Tested residues | BBSome binding: trend from microarray | Validated peptides | Peptide sequences | $K_D$ [µM] |
|---|---|---|---|---|---|---|
| 5-HT-6 | P50406 | 195–440 | 3ICL: weak | not tested | | |
| | | | CT: strong | 5HT6 CT | RDFKRALGRFLPCPRCPRER | 44.2 ± 10.9 |
| MCH-R1 | Q99705 | 288–422 | 3ICL: strong | MCH 3ICL | RILQRMTSSVAPASQRSIRLRTKRVTRT | 68.8 ± 11.3 |
| | | | CT: weak | not tested | | |
| NMU-R1 | Q9HB89 | 243–426 | 3ICL: strong | NMU 3ICL1 NMU 3ICL2 | LLIGLRLRRERLLLMQEAKGRG GRGSAAARSRYTCRLQQHDRGRRQ | 18.1 ± 1.6 >100 |
| | | | CT: weak | not tested | | |
| SSTR3 | P32745 | 44–418 | 3ICL: strong | SSTR3 3ICL | VKVRSAGRRVWAPSCQRRRRSERRVTRM | 12.8 ± 1.07 |
| | | | CT: medium | SSTR3 CT1 SSTR3 CT2 SSTR3 CT3 SSTR3 CT4 SSTR3 CT5 SSTR3 CT6 SSTR3 CT7 SSTR3 CT8 SSTR3 CT9 | RFKQGFRRVLLRPSR GFLSYRFKQGFRRVLLRPSRRVRS GFLSAAFKQGFRRVLLRPSRRVRS GFLSYRAAQGFRRVLLRPSRRVRS GFLSYRFKQGAARVLLRPSRRVRS GFLSAAFKQGAARVLLRPSRRVRS GFLSAAAAQGFRRVLLRPSRRVRS GFLSAAAAQGAARVLLRPSRRVRS GFLSYRFKQGFRRVLLRPSEEVES | 22.1 ± 1.4 0.11 ± 0.02 1.1 ± 0.2 0.40 ± 0.08 1.5 ± 0.3 4.5 ± 1.1 1.0 ± 0.2 11.0 ± 3.3 4.7 ± 1.4 |
| Smo | Q99835 | 234–787 | 3ICL: weak | not tested | | |
| | | | CT: strong | Smo CT1 Smo CT2 Smo CT3 Smo CT4 Smo CT5 Smo CT6 Smo CT7 Smo CT8 Smo (R546-L665) | TLLIWRRTWCRLTGQ KRIKKSKMIAKAFSK TLLIWRRTWCRLTGQSDDEPKRIKKSKMIAKAFSKRH TLLIWRRTWCRLTGQSDDE TLLIAARTWCRLTGQSDDE TLLIWRRTAARLTGQSDDE TLLIAARTAARLTGQSDDE KRLGRKKKRRKRKKE R546-L665 | 1.8 ± 0.5 >100 1.4 ± 0.5 1.1 ± 0.5 9.7 ± 3.3 12.5 ± 3.0 14.3 ± 4.7 6.8 ± 2.5 36.1 ± 3.9 |
| GPR161 | Q8N6U8 | 31–529 | 3ICL: strong | GPR161 3ICL | FIFRVARVKARKVHCG | 6.3 ± 1.5 |
| | | | CT: strong | GPR161 CT1 GPR161 CT2 | NKTVRKELLGMCFGDRYYREPFVQRQR VTARTVPGGGFGGRRGSRTLVS | 67.4 ± 7.3 >100 |

Summary of qualitative and quantitative analysis of binding data for peptides, indicated by their sequence, from those GPCRs for which significant BBSome interactions were found in the peptide microarrays and analyzed for affinity by fluorescence polarization. Basic/aromatic residues in SSTR3 and Smo were mutated to alanine are shown in red, and the corresponding wild type sequences in purple. Previously described CTS sequences are marked in blue (; *Mukhopadhyay et al., 2013*; *Zhang et al., 2012a*).

DOI: https://doi.org/10.7554/eLife.27434.011

3ICL or a (F/Y/W)(K/R) motif in the C-terminal part (*Corbit et al., 2005*). The contribution of binding sites to affinity however varied considerably, with some GPCRs utilizing mostly the third extracellular loop, others the C-terminal region or both (*Table 1*). The interactions with Polycystin-1 correspond to a region of 44 residues that was shown to be important for ciliary targeting and that contains a tyrosine-based sorting signal YEMV (*Su et al., 2015*). The positive Fibrocystin signal, which corresponds to a ciliary targeting sequence previously described (*Follit et al., 2010*), might be a false-positive hit, since the corresponding peptides also interacted with Flag-antibody alone. Curiously, a number of sequences interacting with the Flag antibody were previously described as CTS motifs. However, we cannot exclude that peptides interacting with the control antibody alone may also bind the BBSome. Finally, no interactions were observed with the VxPx ciliary targeting signals from the C-terminus of Polycystin-1 and Rhodopsin or the RVxP motif in the N-terminal domain of Polycystin-2 (*Geng et al., 2006*). Sequence analysis of the binding peptides revealed no unique, single consensus motif for BBSome binding. Instead, we identified multiple binding epitopes with variable affinities for the BBSome (see below).

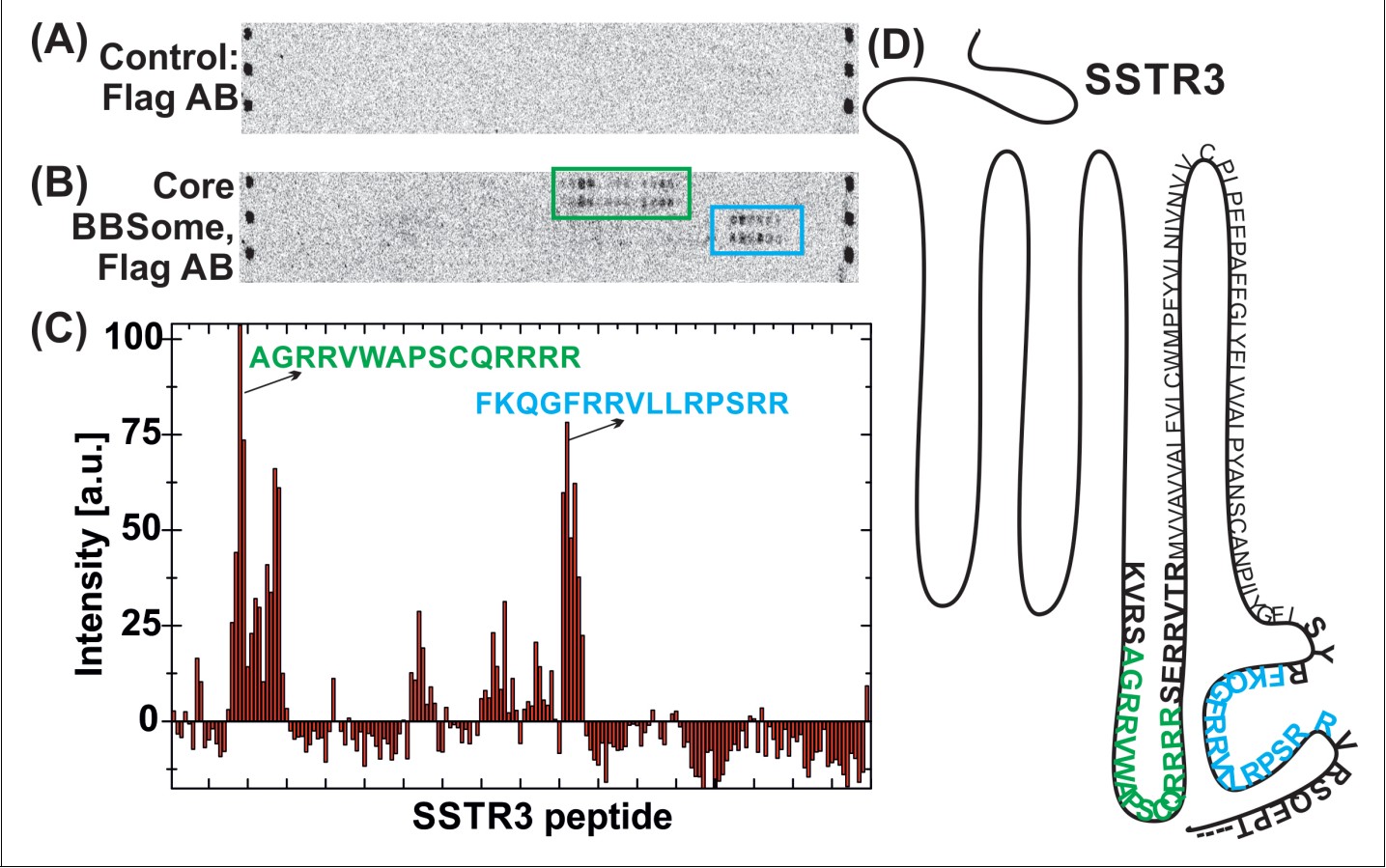

**Figure 4.** Interaction of core BBSome with SSTR3 peptides. Immobilized 15-residue peptides of SSTR3 were incubated with Flag antibody alone (A) or with the core BBSome and subsequently with Flag-antibody (B). The Flag-antibody detected either the Flag-tag on the BBS8 subunit or the Flag control peptides located at the corners of the arrays. (C) Averaged and normalized intensity of the Flag-signal from (B). The positioning of the two strongest peptide hits is presented within a schematic SSTR3 plot (D).

DOI: https://doi.org/10.7554/eLife.27434.009

The following figure supplement is available for figure 4:

**Figure supplement 1.** Peptide microarray with core BBSome.

DOI: https://doi.org/10.7554/eLife.27434.010

## Interaction studies using purified components in solution

To validate and more quantitatively analyze the binding epitopes, we designed fluorophore-labeled peptides based on the results from the peptide microarray and measured their binding to the core BBSome via fluorescence polarization (*Figure 5*, *Figure 5—figure supplement 1*). Our microarray data indicated multiple closely spaced binding epitopes in several cases. Where possible, we combined those epitopes into a single fluorescent peptide with up to 37 residues length to more fully reflect the binding site/motif. *Figure 6* shows a schematic drawing of the tested peptides in relation to previously described ciliary targeting motifs of the respective GPCRs. As summarized in *Table 1*, we determined dissociation constants ($K_D$) of <25 µM for the majority of the tested peptides.

From the 15mer peptides of SSTR3 identified in the microarray screen, SSTR3 CT1 from the C-terminus had the highest affinity (22.1 ± 1.4 µM). Elongation of this peptide by nine residues (SSTR3 CT2), which includes three residues from the 7th transmembrane region and 21 residues from the C-terminus, increased the affinity to 0.11 ± 0.02 µM, the highest of all tested peptides. The peptide SSTR3-3ICL from the third intracellular loop of SSTR3 had an affinity of 12.8 ± 1.0 µM.

In case of Smo, we found interacting peptides only in the C-terminus, in a region that was shown to bind to BBS1 (*Zhang et al., 2012a*). The identified 15mer peptide Smo CT1 and a slightly

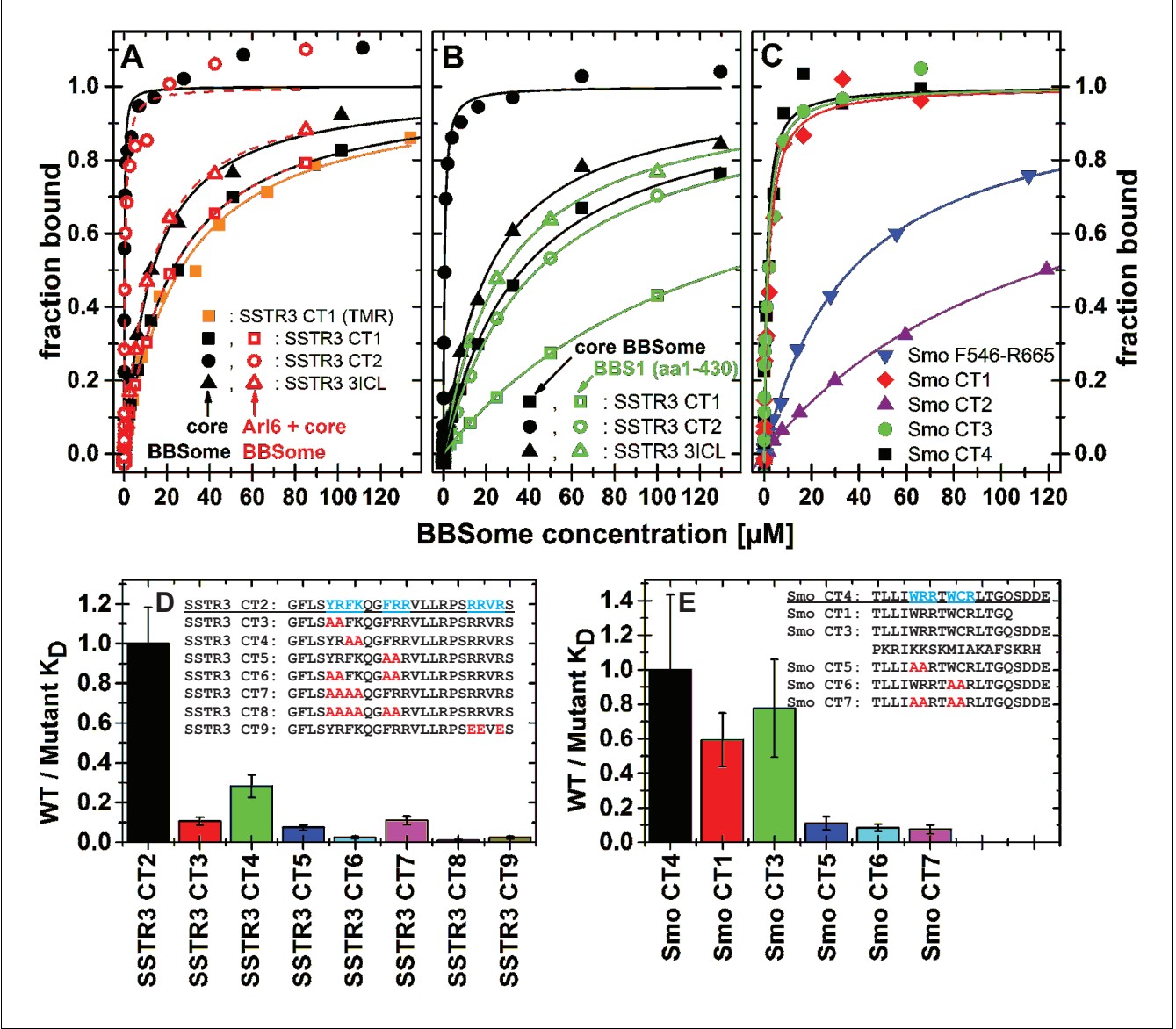

**Figure 5.** Cargo binding to the core BBSome with fluorescence polarization, measured by using fluorescently labeled peptides and the purified core BBSome. (**A,B**) Fluorescent peptides, synthesized as described in methods, from the third intracellular loop of SSTR3 (SSTR3 CT1, SSTR3 CT2) or its C-terminal region (SSTR3 3ICL) were titrated with increasing concentrations of the core BBSome (filled black symbols) or equimolar mixtures of core BBSome with Arl6•GppNHp (open red symbols, **A**) or BBS1 (aa1-430) (open green symbols, **B**). All peptides were N-terminally labeled with fluorescein, with an exception of a C-terminally TMR-labeled SSTR3 CT1 peptide (orange squares) that was used as a control to exclude nonspecific binding events due to the used fluorophore. The increase in fluorescence polarization was fitted to a binding curve as described in the methods section, and the resulting maximal polarization from the fit was defined as 100% of peptide bound to the complex (fraction bound = 1). (Data points exceeding this mark might indicate secondary binding events at high BBSome concentrations). (**C**) Fluorescent peptides from the C-terminus of Smo (Smo CT1-4) and a truncated Smoothened construct (Smo F546-R665) were analyzed as described above. (**D**) The peptide SSTR3 CT2 that bound with highest affinity to the core BBSome was used as a template for the design and analysis of mutant peptides (SSTR3 CT3-9) with substitutions of one or multiple residues as indicated. Polarization experiments were performed as described above and dissociation constants ($K_D$) are presented as ratios to the $K_D$ of the wild type peptide as a bar diagram. (**E**) Bar diagram of the relative affinities of SmoCT1 and CT3 relative to SmoCT4, as determined in C, and of mutant Smo CT4 peptides with substitutions as indicated.

DOI: https://doi.org/10.7554/eLife.27434.012

The following figure supplements are available for figure 5:

**Figure supplement 1.** Cargo binding to the core BBSome.
DOI: https://doi.org/10.7554/eLife.27434.013

**Figure supplement 2.** Mutational analysis of the BBSome binding peptides.

*Figure 5 continued on next page*

Figure 5 continued

DOI: https://doi.org/10.7554/eLife.27434.014

elongated 19mer Smo CT4 bound with affinities of 1.8 ± 0.5 μM and 1.1 ± 0.5 μM, respectively, to core BBSome. Smo CT2, which contains one of two polybasic sequences in the C-terminal cytosolic region of Smoothened, bound only weakly, with an affinity of >100 μM. Consistently, the 37mer peptide Smo CT3 that spans both sequences of SMO CT1 and CT2 had an affinity of 1.4 ± 0.5 μM, which is similar to that of Smo CT1 alone. A second polybasic sequence KRLGRKKKRRKRKKE in the C-terminal region of Smoothened (Smo CT8) had a significant affinity of 6.8 ± 2.5 μM, although it is much lower than Smo CT4. We identified this peptide as a very strong hit in the peptide microarray, but also in the negative control with Flag antibody alone (see above). While the fluorescence polarization experiment indicates a direct interaction with the core BBSome, the results from the peptide microarray suggest a low target specificity, probably caused by the high charge of this motif.

We also tested candidate peptides from the 3ICL and the C-terminal region of GPR161, HTR6, MCHR1 and NMUR1 (summarized in *Table 1*). Of these, the peptides GPR161 3ICL and NMU 3ICL1 had significant affinities of 6.3 ± 1.5 μM and 18.1 ± 1.6 μM, respectively, while the other tested peptides only presented low affinities of >50 μM.

## Mutational analysis of BBSome binding epitopes

Peptides binding to the core BBSome contain multiple aromatic/basic residues, some of which have previously been identified as ciliary targeting motifs (*Figure 5* and *Table 1*). To analyze the influence of individual residues or patches of residues on BBSome affinity, we used the highest affinity peptides SSTR3 CT2 and Smo CT4 as templates to design and analyze peptides in which two or more residues of such epitopes were mutated to alanines. The affinities of these peptides to the core BBSome were then determined by fluorescence polarization and compared with the affinity of the wildtype sequence (*Figure 5*, *Figure 5—figure supplement 2*). For both Smo and SSTR3 we can show that mutations of epitopes with basic and aromatic residues reduce BBSome binding affinity significantly, but that there is no simple universal binding motif at least for these two GPCRs. In the case of SSTR3 we targeted the three epitopes YR, FK and FRR, which were mutated to alanine in various combinations. It appears that the affinity is derived from all three motifs and is somewhat additive. Mutation of two residues from each motif changes affinity between four and fourteenfold, whereas combinations with four or six mutated residues reduce affinity up to ~100 fold. The binding site is even more complex, since mutation of three arginine residues at the C-terminus of peptide SSTR3 CT2, that do not strictly follow a (F/Y/W)(K/R) annotation (RRVR, residues 333–336, mutated to EEVE), also reduces affinity 42 fold.

The Smo CT4 peptide also contains two basic/aromatic motifs. The mutation of either the WRR or the WCR motif changes affinity 9 and 11 fold, respectively. The combined quadruple variant reduces affinity 13 fold, again demonstrating that an extended epitope is recognized by the BBsome.

## Simultaneous binding of cargo and Arl6•GTP

We have shown previously that Arl3 is a ciliary protein that is specifically activated inside cilia (*Lokaj et al., 2015*; *Gotthardt et al., 2015*; *Zhou et al., 2006*). In its GTP-bound form it releases prenylated and myristoylated pepides from the carrier proteins PDE6d and Unc119. We wondered whether the binding of cargo and Arl6•GTP to the BBsome are mutually exclusive, i.e. whether Arl6 has a role in either the binding to or the release of cargo from the BBSome. We thus tested the binding of cargo in the presence of Arl6•GppNHp, but did not find significant Arl6 dependency in our fluorescence polarization assay. By titrating either the core BBSome alone or a stoichiometric mixture of the core BBSome with Arl6•GppNHp to fluorescently labeled peptides SSTR3 CT1, SSTR3 CT2 and SSTR3 3ICL, no significant difference in binding behavior was observed (*Figure 5A*).

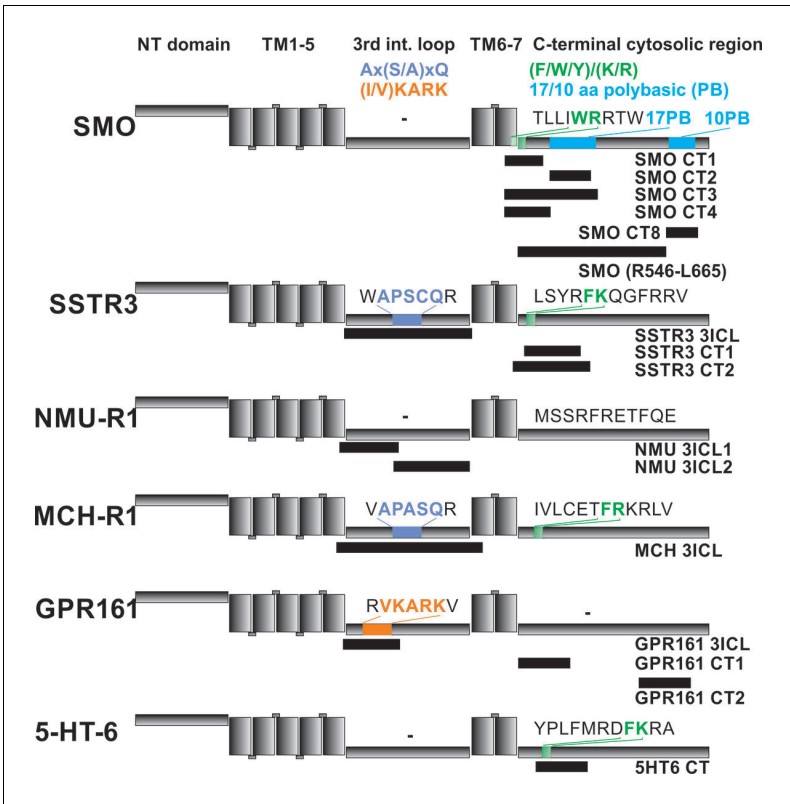

**Figure 6.** Localization of FITC-labeled peptides from various GPCRs used in binding studies and their relation to previously identified CTS motifs. The schematic drawing of the analyzed GPCRs shows the structural elements: NT: N-terminal extracellular domain, TM: transmembrane helices, third intracellular loop and C-terminal cytosolic region). Peptides (for their full sequences see *Table 1*) were designed based on the results of the peptide microarray (*Figure 4* and *Figure 4—figure supplement 1*), and were tested for binding to the core BBSome. Black blocks indicate the approximate position of the peptides relative to the primary sequence, with the corresponding names to the right. The sequences of polybasic regions in Smoothened and of known CTS motifs within the third loop and the C-terminal domain of each GPCR are highlighted in color.

DOI: https://doi.org/10.7554/eLife.27434.015

## Discussion

### The core BBSome complex

Our studies show that multiple BBSome subunits are needed to stabilize each other. An interaction network of the individual BBSome components emerged that is consistent with a recently described immunoprecipitation assay using GFP and RFP fusion proteins (*Katoh et al., 2015*), placing BBS9 as a central organizational unit that forms interactions with BBS1, 2, 5 and 8 and possibly others. We observed that BBS18, despite its small size of ~10 kDa, has a remarkable stabilizing effect on the complex. It is predicted to fold in two α-helices (*Jin et al., 2010*), which might form an elongated structure in the complex that links BBS8, 9 and 4. The core BBSome as the most stable of the complexes analyzed in our studies, composed of BBS 1,4,5,8, 9 and 18, represents the complete set of BBSome subunits for which analogs are found in *Drosophila* (*Nachury et al., 2007*). BBS2 and BBS7 are absent in *Drosophila* and might therefore be dispensable for many aspects of BBSome function. It appears that they are at least dispensable for the formation of a stable monomeric complex that interacts with cargo and Arl6.

### Is the BBSome assembled sequentially or via multiple parallel routes?

The observed formation of a stable core BBSome is in contrast to previous co-immunoprecipitation and co-migration experiments which suggested that BBS6, 10 and 12 are required for BBSome

formation (*Seo et al., 2010*). It has been proposed that the BBSome is assembled from a complex of BBS chaperonines with CCT/TRiC components and that BBS7 acts as a scaffold for further BBSome components to be added in a sequential order (*Zhang et al., 2012b*). This, however, is not consistent with the formation of a stable core BBSome that we observe in the absence of BBS chaperonines, BBS2 and BBS7. All these proteins are also absent in *Drosophila*, indicating that in *Drosophila* no sequential assembly is required. We observed that BBS7 and BBS2, which are highly insoluble proteins by themselves, also reduce the solubility of the entire BBSome complex, in the presence or absence of BBS6, 10 and 12 or any combinations thereof. We propose that the CCT/TRiC complex and the chaperonines might have a role in folding BBS7 and BBS2 before these proteins are added to the BBSome, which might not operate in insect cells. Nevertheless, the formation of the 6mer BBSome indicates that the assembly might follow two parallel routes, one assembling the core BBSome and the second via the BBS/CCT complex to deliver BBS7 and probably also BBS2 to the complex. In conclusion, the insect cell expression machinery seems to be sufficient for folding and assembly of an active core BBSome.

## The core BBSome has a Y-shaped conformation

Although the resolution of the structural model of the 6mer BBSome core is not high enough to localize individual subunits, the Y-shaped arrangement together with the information on binary or oligomeric interactions of BBSome components allows some speculations. The N-terminal ß-propeller domain (residues 1–368) of BBS9 binds to BBS5 and the C-terminal appendage part forms a stable complex with BBS1. We could isolate a stable complex of BBS 4,8,9 and 18, but did not succeed to obtain complexes of BBS1,4,5,8 and 18 in the absence of BBS9. We therefore assume that two of the lobes of the Y contain BBS1 and BBS5, respectively, while the third lobe includes the three remaining proteins BBS4, 8 and 18. BBS9 may hold the assembly together as either a monomer or a dimer. From gel filtration experiments it is clear that BBS9 contains a dimerization module, but the dimerization might or might not persist upon complex formation. Although the gel filtration profile of the core BBSome excludes the presence of a fully dimeric complex, BBS9 could still be present twice in the complex. We are currently working on a high-resolution cryo-EM structure of the 6mer BBSome complex to address its detailed structural arrangement.

## The 6mer BBSome preserves binding to Arl6 and cargo proteins

BBS1 is the most frequently mutated component of the BBSome and is known as a major interaction site that is involved in many contacts with BBSome cargo and other effector proteins (*Zhang et al., 2011*; *Kim et al., 2013*; *Mourão et al., 2014*; *Seo et al., 2009*; *Su et al., 2014*; *Nachury et al., 2007*; *Wei et al., 2012*; *Zhang et al., 2012a*). We could demonstrate that the core BBSome binds to activated GTP-bound Arl6, but not to Arl6•GDP. While the Arl6 interaction only requires the BBS1 ß-propeller (*Mourão et al., 2014*), the interaction with cargo might require the complete BBS1 protein and/or additional BBSome components. Indeed, we were able to identify a small peptide fragment from the C-terminal part of SSTR3 which binds to the core BBSome with ~110 nM affinity, while the affinity to BBS1 (aa1-430) is ~100 fold lower (*Figure 5B*). In contrast, the 3ICL of SSTR3 has similar affinities to the truncated BBS1 subunit and the core BBSome. These findings might indicate that, while the C-terminal domain of SSTR3 requires interactions with BBS1 and other subunits of the core BBSome, the interaction with its 3ICL is mostly mediated by BBS1 alone. The latter is consistent with the previous finding that a GST-tagged 3ICL of SSTR3 binds strongly to BBS1, but less prominently to BBS4,5,8 and 9 (*Jin et al., 2010*). It was also shown that the 3ICL is sufficient to target non-ciliary GPCRs to cilia (*Berbari et al., 2008a*), but that removing this loop from SSTR3 does not prevent ciliary localization. This further supports the importance of the additional high affinity binding epitope in the C-terminal part and clearly indicates the involvement of multiple interaction sites.

For Smoothened, our microarray data suggested no significant binding of the BBSome to its 3ICL, but instead to three sequences within its C-terminal region. One of these contains a (F/Y/W)(K/R) motif C-terminal to the seventh transmembrane domain, and the other two correspond to 17 and 10 residues polybasic stretches. A peptide spanning most of the 17 residue polybasic motif (Smo CT2) presented only weak interaction, while peptides Smo CT3 and CT4 containing both the WRR and WCR motifs showed robust binding (*Table 1*). This finding is consistent with a previous study which indicated BBS1 interaction with the Smo region Q477-L665 containing all three motifs found

in the microarray, but deletion of the 17 residue polybasic motif had minimal effect on BBS1 binding (*Zhang et al., 2012a*). In that study, a deletion of the second, 10 residue polybasic stretch from the C-terminus of Smoothened was found to enhance binding to BBS1, which indicates a possible regulatory role of this region. We found that a peptide containing this motif binds to the core BBSome with reasonable 6.8 ± 2.5 µM affinity, but also that it forms unspecific interactions with the antibody itself, thus demonstrating a low target specificity of this peptide. It was found that neither of the polybasic motifs are strongly affecting the ciliary localization of Smoothened (*Nedelcu, 2013*), supporting the notion that binding to BBSome and ciliary localization are not necessarily overlapping features.

Summarizing these studies, we found robust binding of the core BBSome with certain ciliary targeting signals, particularly with the (F/Y/W)(K/R) motif and other sequences containing one or more basic or aromatic residues. Our mutation data show that a long stretch of residues is required for binding and that several motifs contribute to the affinity. Thus we would assume that the BBSome may contain an extended binding site for target molecules that recognizes general features of aromatic and basic residues rather than a specific sequence. The identified motifs are not exclusive to GPCRs: For example Fibrocystin and Polycystin-2 also contain polybasic sequences that present enhanced BBSome binding, and the C-terminal cytosolic domain of Polycystin-1 contains a (F/Y/W)(K/R) motif located directly carboxy-terminal to its last transmembrane helix, which has enhanced binding to the BBSome (*Figure 4—figure supplement 1*). In contrast, other described CTS like a VxPx motif in the C-terminus of Polycystin-1 and Rhodopsin are not recognized by the BBSome.

Finally we have shown that the binding of cargo and of Arl6•GTP are not mutually exclusive and that the affinities of peptides in fluorescence polarization is not influenced by the presence or absence of the small G protein (*Figure 5*). Thus the function of Arl6 is not analogous to that of Arl3 whose function is the release of cargo from its carriers PDE6d and Unc119.

## Towards a consensus motif for BBSome recognition

While it is well established that the BBSome recognizes proteins containing certain CTS sequences to initiate their ciliary transport, the nature of this recognition and whether or not the interaction sites overlap with CTS sequences are basically unexplored. In our peptide microarray, we found interactions of the core BBSome with regions that are known to contain CTS signals, like the 3ICL and/or the C-terminus of GPCRs, but strong BBSome binding 15mer peptides do not overlap with known CTS sequences in all cases. The (F/Y/W)(K/R) motif is an example of a CTS that overlaps well with our identified binding peptides. It is a hydrophobic and basic residue motif directly carboxy-terminal to the seventh transmembrane helix of GPCRs (*Corbit et al., 2005*; *Deretic, 2006*; *Bhogaraju et al., 2013*). In general we find that arginine residues, particularly when preceded by aromatic residues, are a major factor for strong BBSome interaction. By performing a single-residue walk over the GPCR sequence we can show that enhanced BBSome binding almost invariantly started when the peptide walk approached an arginine residue in the corresponding region. We observed that arginine-rich motifs enhancing BBSome binding are also present in the 3ICL: All tested GPCRs contain an arginine residue close to the N-terminal start of the 3ICL, and peptides containing this arginine have a strong tendency to enhanced BBSome binding. In contrast, we found that CTS sequences that do not contain a clustering of aromatic and basic residues, such as the Ax(S/A)xQ motif (*Bhogaraju et al., 2013*) of MCHR1 and SSTR3 as well as an (I/V)KARK motif from GPCR 161 (*Bhogaraju et al., 2013*; *Mukhopadhyay et al., 2013*) don't seem to be sufficient to bind to the core BBSome: Although we found peptides that overlap with or are in close proximity to these CTS motifs, enhanced binding to the core BBSome does not require the CTS sequence being part of the peptide. This indicates that these CTS motifs may have another function, while core BBSome interaction is mediated, or at least enhanced, by nearby aromatic/basic stretches. We observed that multiple aromatic/basic stretches are additive in enhancing the affinity to the core BBSome, which explains why a single denominator for BBSome interactions is hard to identify. This interplay between different CTS and BBSome interaction sites might further be modulated in the native context of the membrane-inserted GPCR and/or by BBS2 and BBS7, which are missing in the core BBSome.

It should be noted that BBSome binding and ciliary targeting are not necessarily related in a simple fashion. At least for certain proteins the BBSome has been shown to be more important for the export phase of ciliary transport (*Lechtreck et al., 2013*; *Liew et al., 2014*) rather than its import,

indicating that different signals might exist for import, retention or export of BBSome cargo. If we assume that these different signals do not directly influence the physical properties of the BBSome, they might be independent from the BBSome interaction sites. One scenario would be the functioning of CTS as ciliary retention signals, holding the cargo protein within the cilium by interactions that prevent loading or facilitate unloading of the BBSome. This would be in analogy to our previous studies, where we have analyzed the transport of prenylated and myristoylated proteins into cilia. While the affinities to the carrier proteins are a major determinant for entry into the compartment, a ciliary retention signal is additionally required to fully explain the localization of such proteins inside the cilium (*Fansa et al., 2016*). The final localization of a protein would thus be determined by the equilibrium of interactions inside and outside the cilium. Such an equilibrium is dynamic and depends on the signaling properties of the compartment, which is particularly relevant for the Hedgehog pathway which dramatically changes the localization of the Smo and Ptch receptors (*Pusapati and Rohatgi, 2014*). The availability of sufficient amounts of a soluble and stable core BBSome complex will allow a more detailed biochemical and structural analysis of the binding of BBSome to its cargo and the relation of that binding to ciliary transport and retention.

## Materials and methods

### Gene sequences

The sequences of the analyzed BBS proteins comply with the following NCBI reference sequences: BBS1 (NM_024649.4, GI:9029553), BBS2 (NM_031885.3, GI:219842318), BBS3 (NM_032146.3, GI:29826300), BBS4 (NM_033028.3, GI:219842322), BBS5 (NM_152384.2, GI:54145494), BBS6 (BC028973.2, GI:34783542), BBS7 transcript 1 (NM_176824.2, GI:324073523), BBS7 transcript 2 (NM_018190.3, GI:324073529), BBS8 (NM_198309.2, GI:53759117), BBS9 (NM_198428.2, GI:75905799), BBS10 (BC026355.1, GI:20072253), BBS12 (NM_001178007.1, GI:295821197, with five variations G1157A, G1380C, G1399A, C1410T, A1872G), BBS18 (NM_001195306.1, GI:305855073).

### Construct design and cloning

The genes encoding for human BBS subunits BBS1-10 and BBS12 and also the tested heterologously expressed Smoothened fragments were cloned into vectors from the ACEMBL system for multiprotein complex expression (*Vijayachandran et al., 2011*). Since all of these genes have in common the lack of restriction sites RsrII and SalI, we amplified them by PCR with appropriate primers to introduce an RsrII restriction site at the 5′ end (N-terminal end of the encoded protein) and a SalI site at the 3' end. PCR fragments generated this way could be directly integrated in the pIDC donor vector by classical restriction site cloning to encode for the untagged proteins. The acceptor vector pACE-Bac1 was modified to accommodate RsrII and SalI sites as well as N-terminal tags (His6 tag, Flag-tag, Strep-tag or a dual-Strep tag with additional Prescission protease cleavage site) optimally positioned for gene insertion. The full modified and linearized vectors were generated this way by PCR from the original vectors as template, allowing subsequent DpnI digestion to completely remove the template vector. The following primer pairs were used for this:

pACEBac1 with His6-tag (5′-CACTGAGTCGACGAGCTCACTTGTCG-3′ and 5′-AGCATCGGACCGTGATGGTGATGGTGATGCATGGATCCGCGCCCGATGGTGGGACGGTATG-3′), pACE-Bac1 with Flag-tag (5′-CACTGAGTCGACGAGCTCACTTGTCG-3′ and 5′-AGCATCGGACCGGCCTTATCGTCGTCATCCTTGTAATCCATGGATCCGCGCCCGATGGTGGGACGGTATG-3′), pACEBac1 with Strep-tag (5′-CACTGAGTCGACGAGCTCACTTGTCG-3′ and 5′-AGCATCGGACCGGCTTTTTCGAACTGCGGGTGGCTCCAGCTAGCCATGGATCCGCGCCCGATGGTGGGACGGTATG-3′.

The dual-Strep-tagged pACEBac1 with included Prescission protease cleavage sequence was made with the previously generated Strep-tagged pACEBac1 as template. To improve PCR efficiency we utilized a PCR strategy with three primers: (5′-CACTGAGTCGACGAGCTCACTTGTCG-3′, 5′-GAGAGGACCTTGAAACAGCACTTCGAGTCCGGCGCCTTTCTCGAACTGTGGGTGGGACCAAC-CAGCACCACTTCCTGCCGCTGATCCGGCAGAGCCCTTCTCGAACTGGGGATGGCTCCA-3′ and 5′-GAGAGGACCTTGAAACAGCACTTCG-3′). The last primer binds to the end of amplification products of the second primer. The two shorter primers were used in 10-fold excess over the long primer, this way reducing the risk of unspecific amplification. The generated linearized vector contains SalI

and EcoO109I restriction sites instead of SalI and RsrII. The EcoO109I sequence that we used also encodes for the last residues of Prescission protease. Upon cleavage with EcoO109I, a sticky end that is compatible with RsrII is generated, allowing a linker-free fusion with the desired genes. The full protein sequence that this way is added N-terminally to a fused gene is MASWSHPQFEKGSAG SAAGSGAGWSHPQFEKGAGLEVLFQGP.

This strategy generated a combinatorial toolbox with a set of vectors including the desired N-terminal tags and the genes with restriction sites for insertion into these vectors in frame with the tags. By ligation of any combination of RsrII/EcoO109I and SalI digested vectors and gene fragments from this set we could generate vectors encoding for genes which were either not tagged (by integration in the pIDC vector) or that contained the desired tag at the N-terminus (by integration in the appropriately modified pACEBACI vector). C-terminal tags were not introduced into the linearized vector fragments, but were added to the gene fragments by appropriate PCR primers. These were then combined with vectors designed as described above without additional N-terminal tags. For multigene combinations, we first combined multiple expression cassettes (containing promoter, gene and terminator sequence) from the generated acceptor and donor vectors utilizing the homing endonuclease sites that are present for this purpose in the ACEMBL system (*Vijayachandran et al., 2011*). This way we generated vectors with up to four expression cassettes (coding for four different BBS genes). Even larger multigene vectors were generated by a subsequent Cre-LoxP combination of the integrated LoxP sequences on each acceptor and donor vector.

## Generation of baculovirus

The baculovirus was generated according to the manufacturer's protocol. Briefly, the generated ACEMBL acceptor vectors (or multigene constructs with an integrated acceptor vector) were transformed into chemically competent *e.coli* DH10 EmBacY cells carrying an EmbacY bacmid. The site-specific transposition that integrates the desired gene constructs into the bacmid by a target-specic Tn7 transposon also interrupted a LacZ reporter gene, which was verified by blue-white screening. Colonies with the recombinant bacmid were amplified in LB medium and subjected to bacmid purification. 10 µg of the purified bacmid and 4 µL FuGENE HD DNA transfection reagent (Promega, Germany) were used to transfect 1 mL Sf9 insect cells with a density of 1E6 cells/ml in SF900 III serum free medium (Thermofisher Scientific, Waltram, MA). The cells were incubated at 27.5°C for 3 days and the whole cell suspension was used to infect another 10 ml of SF9 insect cells with a density of 1E6 cells/ml for 4 days. 10% FBS (Thermofisher Scientific) were added to the cell supernatant and the generated 'V1' virus was used either directly to infect expression cultures or to infect more SF9 cells, to generate larger volumes of 'V2' virus.

## Protein expression and purification

Proteins and protein complexes were overexpressed in Hi5 insect cells (Thermofisher Scientific) as follows: Hi5 cells at a density of 1E6 cells/ml in SF900 III medium were infected with V1 or V2 virus stocks. A virus dilution of 1:100-1:500 was used and adjusted to stop cell division after reaching a density of 1.6–1.8 E6 cells/ml. Cells were shaken at 120 rpm and 27.5°C for 65 hr post-infection, harvested by centrifugation at 3000 rpm, and resuspended in lysis buffer (50 mM Hepes pH 8.0, 150 mM NaCl, 5 mM $MgCl_2$, 10% glycerol, 1 mM PMSF, 1 mM benzamidine and reducing agent (5 mM ß-mercaptoethanol (ß-ME), 1 mM DTT or 1 mM TCEP depending on the requirements of the subsequent purification steps). The resuspended cells were either immediately lysed by douncing with a Dounce Tissue Grinder (Wheaton, Millville, NJ) 30–40 times, or frozen at −20°C and then dounced 20 times after thawing. The cell debris was removed by centrifugation at 25000 rpm for 60 min, and the supernatant was used for further protein purification.

For affinity purifications based on Strep- or Dual-Strep-tagged proteins, the cell supernatant was mixed with 'BioLock' Biotin blocking solution (IBA, Germany) according to the manufacturers protocol and then loaded on a column filled with Strep-Tactin Superflow high capacity resin (IBA). The column was washed with 3 CV washing buffer (50 mM Hepes pH 8.0, 150 mM NaCl, 5 mM $MgCl_2$, and 10% glycerol) supplemented with 5 mM ß-ME. Subsequent washes with 3 CV gel filtration buffer supplemented with 1 mM ATP and 150 mM KCl and then 2 CV washing buffer removed further impurities and chaperones. Then the proteins were eluted with washing buffer supplemented with 10 mM D-desthiobiotin (IBA). Histidine-tagged protein complexes are further purified with $Ni^{2+}$–

NTA (Qiagen, Germany) affinity resin and Flag-tagged complexes with anti-Flag M2 affinity resin (Sigma, Germany). The elutions from Strep affinity purifications were directly loaded on the respective column. $Ni^{2+}$-NTA-columns were washed with washing buffer supplemented with 15 mM imidazole and 5 mM ß-ME and eluted with washing buffer with added 250 mM imidazole and 5 mM ß-ME. Anti-Flag M2 affinity columns were washed with washing buffer and eluted with washing buffer supplemented with 0.2 mM Flag peptide (sequence DYKDDDDK, Proteogenix, France). If required, dual-Strep tags were removed by incubation with Prescission protease at 1:100 stoichiometric ratios for 12 hr.

Affinity-purified proteins were subjected to size-exclusion chromatography on a Superdex 200 10/300 column or on a Superose 6 5/150 column (GE Healthcare, Germany) in gel filtration buffer (50 mM Hepes pH 8.0, 150 mM NaCl, 5 mM $MgCl_2$, 10% glycerol and 1 mM TCEP). For small-scale purifications, a Superdex 200 5/200 column was manually prepared from Superdex 200 material with 13 µm particle size (GE Healthcare).

## Negative stain electron microscopy

Freshly prepared BBSome (6-mer) after gel filtration chromatography was examined by negative stain electron microscopy. Briefly, 4 µl of sample were absorbed on freshly glow-discharged carbon coated copper grids for 1 min at room temperature. Excess of sample was removed by blotting with filter paper (Whatman no 4). Grids were washed twice with gel filtration buffer and stained with 0.75% uranyl formate for 1 min, blotted and air-dried. Grids were examined and imaged at a JEOL 1400 microscope equipped with a $LaB_6$ cathode operating at 120 kV at a magnification of 60,000 (Pixel size 1.89 Å). Digital micrographs were recorded at low dose conditions using a 4k × 4 k CMOS camera F416 (TVIPS, GmbH).

## 2-D alignment and 3-D reconstruction

19,350 particles were selected using e2boxer (*Tang et al., 2007*). Alignment and classification was executed using reference-free alignment and the ISAC classification procedure incorporated in SPARX and EMAN2 (*Hohn et al., 2007*; *Tang et al., 2007*). The best 140 class averages were selected and used for an *ab initio* 3-D reconstruction using sxviper (SPARX). The *ab initio* model was used as reference for the final 3-D reconstruction using 8339 raw particle images with sxali3d (SPARX). The final resolution of the reconstructed model was 23 Å, as estimated by Fourier shell correlation (criterion of 0.5). All further analysis, visualization and EM figures were done in Chimera (*Pettersen et al., 2004*).

## Pulldown assays

Pulldown assays were performed using 20 µl Strep-Tactin Superflow high capacity resin (IBA, Germany) per assay equilibrated with pulldown buffer (50 mM Hepes pH 8.0, 150 mM NaCl, 5 mM $MgCl_2$, 5 mM ß-ME and 0.05% DDM). Smo(R546-L665, N-terminal Strep-tag) and Arl6 (residues 17–186, Q73L, C-terminal Strep-tag, with the nucleotide exchanged to either GDP or GppNHp) was immobilized as the bait, and the 6mer BBSome core complex served as the prey (The dual-Strep-tag on BBS18 within the complex was completely removed by Prescission protease cleavage for this experiment). 35 µg of the bait proteins were incubated with the beads in 1 ml pulldown buffer at 4°C for 1 hr, and the unbound protein was washed away with 3 × 1 ml pulldown buffer. The resin was then incubated with 35 µg (Smo(R546-L665)) or 75 µg (6mer BBSome core) of the prey proteins for 1 hr at 4°C and unbound protein was washed away with four additional washing steps. Proteins were then released from the resin with pulldown buffer containing 10 mM D-Desthiobiotin (IBA).

## Peptides

Fluorescently labeled peptides were purchased from Proteogenix (Proteogenix SAS, France). Fluorescein-labels are located N-terminally to the peptide sequence and TAMRA at the C-terminal side. The full sequence of the tested peptides is listed in *Table 1*.

## Fluorescence polarization measurements

Fluorescence polarization experiments were performed using a Fluoromax-4 spectrophotometer (HORIBA Jobin Yvon, Germany) or a Safire two multimode plate reader (Tecan, Switzerland).

Fluoresceine-labeled samples were measured with excitation/emission wavelengths of $\lambda_{Ex}$ = 490 nm and $\lambda_{Em}$ = 520 nm with a bandwidth of 7 nm on the Fluoromax-4. On the plate reader, $\lambda_{Ex}$ = 470 nm and $\lambda_{Em}$ = 525 nm with an emission bandwidth of 5 nm was used. TAMRA-labeled peptides were detected at $\lambda_{Ex}$ = 550 nm and $\lambda_{Em}$ = 580 nm with a bandwidth of 5 nm on the Fluoromax-4. Sample volumes were 20 µl in a 1,5 × 1,5 mm cuvette (Hellma Analytics, Germany) for the Fluoromax-4 and 4.5 µl in a 384 well low-binding plate (Corning, Germany). Data analysis was done with the Origin 9.0 program suite (Originlab, Northampton, MA). The polarization curves were fit against the function

$$F_I = F_{min} - (F_{min}-F_{max}) \times (E + L + K_D - \sqrt{(E + L + K_D)^2 - 4 \times E \times L}) / (2 \times E)$$

with $F_I$, $F_{min}$ and $F_{max}$ corresponding to the fluorescence polarization signal and the signal of unbound and completely saturated fluorescent molecule E with interaction partner L, respectively. The concentrations of molecule E are indicated in the corresponding figure legends, and the calculated dissociation constants $K_D$ are listed in *Table 1*.

## Peptide microarrays

Peptides from potential BBSome cargo proteins were purchased from PEPperPRINT (Germany) as 15-residue peptides immobilized on a microscope slide (*Beyer et al., 2007*; *Stadler et al., 2008*). Each slide contained two arrays of 60 × 68 positions. From each array, two rows are used for control peptides that directly bind to tag-specific antibodies, every second of them being a Flag and an HA sequence. In the presented experiment, only the Flag sequences are detected by the used Flag antibody (anti-Flag M2, Dylight 800; PEPperPRINT). The microarray slide was positioned in an incubation chamber (PEPperPRINT), which separates the two arrays from each other and allows the individual incubation of each array with target proteins. The arrays were first blocked with BSA according to the manufacturer´s protocol. Next, unspecific binding of Flag antibody to the immobilized peptides was tested: 1 µl of Flag antibody in 1 ml array buffer (50 mM Hepes pH 8.0, 150 mM NaCl, 5 mM MgCl$_2$, 0.05% Tween20) supplemented with 0.1% fat-free BSA was incubated with the array for 30 min. The assay slide was washed 3 times with 1 ml of array buffer, dipped in 1 mM Hepes pH 8.0 and dried. The fluorophore-labeled antibody was detected on a fluorescence plate reader Odyssey Clx (LI-COR, Germany) at 800 nm. Afterwards, the array was incubated with 150 µg of the 6mer BBSome core complex in 1 ml array buffer supplemented with 0.1% fat-free BSA over night at 4°C on a shaking platform, washed 3 times with 1 ml array buffer, and Flag-tagged BBS8 from the complex was detected by a subsequent incubation with Flag antibody as described for the antibody control.

## Accession codes

Coordinates of the EM structure have been deposited in the Electron Microscopy Data Bank under accession code EMD-3712.

## Acknowledgements

We thank Dr. Christos Gatsogiannis for his guidance for EM processing and Jana Seidel for technical assistance. This work was funded by the Max Planck Society (to SR and AW), the European Research Council under the European Union's Seventh Framework Program (FP7/2007-2013) (grant number 615984 to SR), the European Research Council (grant number 268782 to AW), and the Sonderforschungsbereich-DFG (SFB 642) (to AW).

## Additional information

### Funding

| Funder | Grant reference number | Author |
| --- | --- | --- |
| European Commission | ERC 615984 | Stefan Raunser |
| Max-Planck-Gesellschaft | Open-access funding | Stefan Raunser<br>Alfred Wittinghofer |
| European Commission | ERC 268782 | Alfred Wittinghofer |

| Deutsche Forschungsge-meinschaft | SFB 642 | Alfred Wittinghofer |

The funders had no role in study design, data collection and interpretation, or the decision to submit the work for publication.

## Author contributions
Björn Udo Klink, Conceptualization, Data curation, Formal analysis, Supervision, Writing—original draft, Project administration, Writing—review and editing, Methodology, Investigation, Visualization, Validation; Eldar Zent, Conceptualization, Formal analysis, Writing—review and editing, Validation; Puneet Juneja, Formal analysis, Investigation, Visualization, Validation, Writing—review and editing; Anne Kuhlee, Formal analysis, Investigation, Visualization; Stefan Raunser, Resources, Supervision, Funding aquisition, Writing- review and editing; Alfred Wittinghofer, Conceptualization, Data curation, Resources, Supervision, Funding acquisition, Writing—original draft, Project administration, Writing—review and editing, Validation

## Author ORCIDs
Björn Udo Klink (iD) http://orcid.org/0000-0002-0946-4456
Stefan Raunser (iD) https://orcid.org/0000-0001-9373-3016
Alfred Wittinghofer (iD) http://orcid.org/0000-0002-5800-0236

## Decision letter and Author response
Decision letter https://doi.org/10.7554/eLife.27434.017
Author response https://doi.org/10.7554/eLife.27434.018

# Additional files
## Supplementary files
• Transparent reporting form
DOI: https://doi.org/10.7554/eLife.27434.016

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
