## [Decision Letter]

Thank you for submitting your article "A recombinant BBSome core complex and how it interacts with ciliary cargo" for consideration by *eLife*. Your article has been favorably evaluated by Fiona Watt (Senior Editor) and three reviewers, one of whom, Suzanne Pfeffer, is a member of our Board of Reviewing Editors. The following individual involved in review of your submission has agreed to reveal their identity: Esben Lorentzen (Reviewer #2).

The reviewers have discussed the reviews with one another and the Reviewing Editor has drafted this decision to help you prepare a revised submission.

This is a high quality manuscript that characterizes the BBsome core complex after expression in Baculovirus infected insect cells, reports a low resolution negative stain EM structure, and explores the broad substrate specificity for cytoplasmic peptides representing G protein coupled receptor cytoplasmic domains that rely on the BBsome for ciliary targeting. Through hierarchical complex subunit expression, the authors show that the BBsome exists as a BBS9 seeded structure with BBS 1, 4, 5, 8, and 18 tightly attached and BBS 2 and 7 less so, and this core bound Arl6 in a GTP-dependent manner. Using 4000, 15mer peptide scanning, they identify key interaction motifs in GPCRs and report their interaction affinities using fluorescence polarization – no change in binding affinity is seen in the presence of Arl6-GTP. Although findings will be of broad interest to those studying ciliary protein targeting, the reviewers felt strongly that some physiological test of the sequences identified (or higher resolution structure) would greatly enhance the significance of the present story. Specific suggestions follow here.

Essential revisions:

The authors need to include a detailed analysis of one putative BBS-binding and cilia-localization motif. The "F/W-K/R" motif is a particularly good one since it is cited in countless reviews and is a source of considerable confusion in the field. In addition to mutagenesis and in vitro Bbsome binding assays, which should be quite straightforward, it is important that they make the relevant mutation in intact receptors and show that there is a defect in ciliary trafficking and not only in folding/stability. The assays here are not too difficult, as they involve transient expression of mutants and tests for surface expression and cilia localization (by IF) and ER exit (by EndoH sensitivity) (see below).

Additionally, we recommend that the authors focus on receptors other than Smo. There is a publicly available PhD thesis by Daniel Nedelcu from Adrian Salic's lab that reports a very thorough mutagenesis of the SMO tail with both cilia-localization and signaling assays. The salient result is ciliary localization of SMO is unaffected by both mutations in the "WR" motif or in the polybasic stretch suggested by the authors as important for Bbsome binding. (Interestingly, the polybasic mutation reduces signaling without affecting ciliary accumulation). Thus, it’s not clear that the interaction assays used by the authors have much relevance for the ciliary trafficking of Smo.

Detailed comments:

1) In the Introduction (fourth paragraph), the authors write that the function of the BBSome is import of ciliary membrane proteins such as SSTR3 and components of the hedgehog pathway. Although this was the consensus in the field some years back, this view may not be correct and appears not to be supported by the more recent literature. There is, however, strong data supporting a role for the BBSome in the ciliary export of membrane proteins. Data from Karl Lechtrech using *Chlamydomonas* showed that PLD requires the BBSome for ciliary export and accumulates in the cilium in BBS mutants. Several papers from Greg Pazours lab have demonstrated that the BBSome together with IFT25/27 are required for the export of hedgehog components and the latest 2017 paper from the Nachury lab explains why the BBSome was initially falsely believed to be involved in the import rather than export of ciliary GPCRs. Indeed, mutation of the BBSome machinery results in accumulation of SSTR3 followed by shedding in ectosomes rather than failure to import of SSTR3. Along the same lines, the authors do not once mention TULP3, which is likely the true import adaptor for many ciliary GPCRs. Please update the manuscript with references to the work of Saikat Mukhopadhyay on IFT-A/TULP3 and try to give a more balanced account of the role of BBSomes and TULP3 in ciliary trafficking.

2) Figure 1. The presentation of the gel in 1A is not very nice. It is hard to make sense of what is what. Did BBS18 run out in the last 3 lanes? Or is it the band running much higher than BBS18 in lane 5 (in which case you cannot use the same marker for all) ?

3) One caveat of the polarization experiment is that the BBSome complex used is missing two subunits and that the peptides of ciliary cargoes are not in the context of the native proteins, which could make a significant difference for the C-terminal peptides that contain residues from the last trans-membrane helix. Make sure this is explicitly stated in the manuscript.

4) One conclusion from the manuscript is that '(F/W)(K/R)' containing peptides from the C-terminal region of cargo proteins bind BBSomes with relatively high affinity. However, these peptide sequences often contain multiple similar sequences (FK and FR in SSTR3; WR and WCR in Smo). The authors should mutate each of these regions and determine affinities to try to narrow down the minimal CTS. It would of course be optimal to do this experiment in the context of the full length proteins but that would probably be too much to ask.

5) Motifs in SSTR3 that bind to the BBSome in vitro: The third intracellular loop of SSTR3 has been shown to bind to the BBsome previously. It is sufficient to mediate ciliary targeting but its deletion does not abrogate SSTR3 localization in cilia. The authors identify an additional BBSome interacting motif in the receptor tail using their in vitro assays. However, no tests of specificity are shown – are there residues in this motif that when mutated prevent binding to the BBSome and also prevent ciliary localization of SSTR3 in cells?

6) BBSome Motifs in Gpr161 are identified but their physiological relevance for cilia trafficking or signaling is not investigated. Gpr161 is a negative regulator of basal Hedgehog signaling that is localized in the ciliary membrane at baseline. Hh ligands drive the clearance of Gpr161 from cilia, thought to be a key event in signal activation. Thus, ciliary localization, delocalization and Hh signaling can all be easily assayed after making mutations in the peptide motifs identified by the in vitro BBSome binding assays. Do mutations that abrogate binding to the BBSome also impair any of these physiological activities of the receptor?

7) The authors suggest that the (F/W/Y)-(R/K) motif is discussed extensively as a "general motif" important motif for BBSome recognition. However, there is no functional evidence for this assertion. This motif is present in many GPCRs and there is no evidence that it is particularly enriched in ciliary GPCRs. Second, there is little evidence that mutation of this motif leads to a defect in ciliary trafficking (as opposed to a folding defect that leads to ER retention). This scenario is likely to be the case for this motif in Smo, because mutation of this motif in SMO leads to only a minor effect on signaling and ciliary localization (see PMIDs 16459297, 21474452, and the PhD thesis https://dash.harvard.edu/bitstream/handle/1/11181143/Nedelcu_gsas.harvard_0084L_11080.pdf?sequence=1). Indeed, this same "WR" motif is present in *Drosophila* SMO, which does not localize in cilia!

---

## [Author Response]

Essential revisions:The authors need to include a detailed analysis of one putative BBS-binding and cilia-localization motif. The "F/W-K/R" motif is a particularly good one since it is cited in countless reviews and is a source of considerable confusion in the field. In addition to mutagenesis and in vitro Bbsome binding assays, which should be quite straightforward, it is important that they make the relevant mutation in intact receptors and show that there is a defect in ciliary trafficking and not only in folding/stability. The assays here are not too difficult, as they involve transient expression of mutants and tests for surface expression and cilia localization (by IF) and ER exit (by EndoH sensitivity) (see below).

As suggested, we included a detailed mutagenesis study of actually two peptides, using those with highest affinity. These are derived from the C-terminal regions following the 7TMR domain of SSTR3 and of Smoothened, and both include multiple aromatic/basic sequences, which partially follow the F/W/Y-K/R annotation. We updated Figure 4 with two additional panels (D, E). Please note that we also moved it a bit down within the manuscript for clarity, so that Figure 4 now is called Figure 5. In brief, our mutational analysis indicates that multiple aromatic/basic motifs act together to establish high affinity towards the BBSome.

Additionally, we recommend that the authors focus on receptors other than Smo. There is a publicly available PhD thesis by Daniel Nedelcu from Adrian Salic's lab that reports a very thorough mutagenesis of the SMO tail with both cilia-localization and signaling assays. The salient result is ciliary localization of SMO is unaffected by both mutations in the "WR" motif or in the polybasic stretch suggested by the authors as important for Bbsome binding. (Interestingly, the polybasic mutation reduces signaling without affecting ciliary accumulation). Thus, it’s not clear that the interaction assays used by the authors have much relevance for the ciliary trafficking of Smo.

Comment/modification of the manuscript:

We do not claim that BBSome binding and ciliary targeting are always related in a simple fashion, to the contrary. We discuss in the text that BBSome binding, ciliary localization and export might well not be overlapping functions. We also discuss the analysis of Daniel Nedelcu of the Smo tail. He showed that multiple redundant CTS motifs exist, and found that ciliary localization and signaling are related to each other in a nontrivial fashion. Likewise, cargo binding to the BBSome might or might not be related to ciliary targeting and/or signaling in a more complicated way than anticipated since the localization of Smo is dynamic and dependent on the signaling state of the cell. To unravel the underlying mechanisms, we performed a detailed analysis of the involved binding epitopes for BBSome binding and made mutations of the motifs. As an example, the described WRR motif in mouse Smo tail indeed has an analogous WKR motif in *Drosophila* Smo. However, there is a second WCR motif that is missing in *Drosophila* which does not strictly follow the F/W-K/R notion but according to our mutational analysis is equally important for BBSome binding (compare our newly added Figure 5).

We observed that one of the polybasic motifs in the Smo tail has relatively low affinity towards the BBSome and could be a candidate of a motif that is more relevant to function and/or ciliary retention. We also found that a second polybasic motif has higher affinity but also binds unspecifically to Flag antibody, indicating a low target specificity.

Although it seems obvious that efficient BBSome-dependent transport will require a substantial affinity of the cargo protein to the BBSome, it is also probable that some of the BBSome binding epitopes in Smo have regulatory roles that are not yet fully understood.

Detailed comments:1) In the Introduction (fourth paragraph), the authors write that the function of the BBSome is import of ciliary membrane proteins such as SSTR3 and components of the hedgehog pathway. Although this was the consensus in the field some years back, this view may not be correct and appears not to be supported by the more recent literature. There is, however, strong data supporting a role for the BBSome in the ciliary export of membrane proteins. Data from Karl Lechtrech using Chlamydomonas showed that PLD requires the BBSome for ciliary export and accumulates in the cilium in BBS mutants. Several papers from Greg Pazours lab have demonstrated that the BBSome together with IFT25/27 are required for the export of hedgehog components and the latest 2017 paper from the Nachury lab explains why the BBSome was initially falsely believed to be involved in the import rather than export of ciliary GPCRs. Indeed, mutation of the BBSome machinery results in accumulation of SSTR3 followed by shedding in ectosomes rather than failure to import of SSTR3. Along the same lines, the authors do not once mention TULP3, which is likely the true import adaptor for many ciliary GPCRs. Please update the manuscript with references to the work of Saikat Mukhopadhyay on IFT-A/TULP3 and try to give a more balanced account of the role of BBSomes and TULP3 in ciliary trafficking.

We modified the text to reflect the more recent literature about TULP3 and about the role of the BBSome in export rather than import.

2) Figure 1. The presentation of the gel in 1A is not very nice. It is hard to make sense of what is what. Did BBS18 run out in the last 3 lanes? Or is it the band running much higher than BBS18 in lane 5 (in which case you cannot use the same marker for all) ?

We updated the misleading representation in Figure 1, and added some more annotation for clarity. Actually both BBS18 bands in lane 5 and in the last 3 lanes are correct, but the constructs in the last three lanes contain an uncleaved twin-Strep on BBS18, while the tag is cleaved off in lane 5, which leads to a remarkable change in gel mobility (also compare Figure 1—figure supplement 3).

3) One caveat of the polarization experiment is that the BBSome complex used is missing two subunits and that the peptides of ciliary cargoes are not in the context of the native proteins, which could make a significant difference for the C-terminal peptides that contain residues from the last trans-membrane helix. Make sure this is explicitly stated in the manuscript.

We included a notion on in the first paragraph of the subsection “Identification of binding epitopes in BBSome cargo proteins”, which discusses the limitations of our approach.

4) One conclusion from the manuscript is that '(F/W)(K/R)' containing peptides from the C-terminal region of cargo proteins bind BBSomes with relatively high affinity. However, these peptide sequences often contain multiple similar sequences (FK and FR in SSTR3; WR and WCR in Smo). The authors should mutate each of these regions and determine affinities to try to narrow down the minimal CTS. It would of course be optimal to do this experiment in the context of the full length proteins but that would probably be too much to ask.

A detailed mutational analysis of the multiple similar sequences in F/W-K/R containing peptides are presented now in Figure 5 and Table 1 (YR, FK and FR and also a short arginine-rich sequence RRVR in the SSTR3 CT2 peptide; WR and WCR in the Smo CT4 peptide). Consistent with our findings from the peptide microarray, the binding affinity cannot be narrowed down to any single one of these aromatic/basic motifs. Instead, all of them contribute to BBSome affinity, and mutation of multiple motifs results in a cumulatively reduced binding affinity.

5) Motifs in SSTR3 that bind to the BBSome in vitro: The third intracellular loop of SSTR3 has been shown to bind to the BBsome previously. It is sufficient to mediate ciliary targeting but its deletion does not abrogate SSTR3 localization in cilia. The authors identify an additional BBSome interacting motif in the receptor tail using their in vitro assays. However, no tests of specificity are shown – are there residues in this motif that when mutated prevent binding to the BBSome and also prevent ciliary localization of SSTR3 in cells?

As already described in comment 4), we added a detailed mutational analysis of this motif to the paper. In brief, the conclusion is that each of the four analyzed aromatic/basic motifs in the best-binding peptide from SSTR3 CT contribute to BBSome binding, with the “second” motif (which is discussed in the work of Corbit et at. 2005 as the conserved ciliary localization sequence of SSTR3) being least important.

6) BBSome Motifs in Gpr161 are identified but their physiological relevance for cilia trafficking or signaling is not investigated. Gpr161 is a negative regulator of basal Hedgehog signaling that is localized in the ciliary membrane at baseline. Hh ligands drive the clearance of Gpr161 from cilia, thought to be a key event in signal activation. Thus, ciliary localization, delocalization and Hh signaling can all be easily assayed after making mutations in the peptide motifs identified by the in vitro BBSome binding assays. Do mutations that abrogate binding to the BBSome also impair any of these physiological activities of the receptor?

As stated above, we do not probe the physiological activity of GPR161 in this work. It should be noted that in the first version of this work we were only able to validate GPR161 peptides with relatively low BBSome affinity, which would further complicate such a study. However, we were now able to find a peptide of the 3ICL of GPR161 which is soluble (in contrast to the one we analyzed before), and presents a moderate BBSome affinity. We updated Table 1 accordingly. This peptide might be the best target for future in vivo experiments.

7) The authors suggest that the (F/W/Y)-(R/K) motif is discussed extensively as a "general motif" important motif for BBSome recognition. However, there is no functional evidence for this assertion. This motif is present in many GPCRs and there is no evidence that it is particularly enriched in ciliary GPCRs. Second, there is little evidence that mutation of this motif leads to a defect in ciliary trafficking (as opposed to a folding defect that leads to ER retention). This scenario is likely to be the case for this motif in Smo, because mutation of this motif in SMO leads to only a minor effect on signaling and ciliary localization (see PMIDs 16459297, 21474452, and the PhD thesis https://dash.harvard.edu/bitstream/handle/1/11181143/Nedelcu_gsas.harvard_0084L_11080.pdf?sequence=1). Indeed, this same "WR" motif is present in Drosophila SMO, which does not localize in cilia!

With the new mutational analysis of the (F/W/Y)-(R/K) containing peptides of Smo and SSTR3 that we include in the manuscript (Figure 5), we now have evidence that such motifs positively affect BBSome binding affinity. However, it is also clear that one such motif is not sufficient. For example, the motif that is conserved in *Drosophila* is WKR, compared to a more favorable WRRTWCR in mSmo. Our mutational analysis indicates that both a WRR =>AAR and a WCR=>AAR mutation in the latter both dramatically reduces (but does not eliminate) BBSome affinity. In brief, our data suggest that an interplay of multiple BBSome-binding motifs is required for a high affinity, and that the most relevant BBSome-binding motifs (at least for interactions with the core BBSome) are aromatic/basic sequences.